# Stock Reduction Analysis of Bigeye Croaker *Micropogonias megalops* in the Upper Gulf of California, Mexico

Ricardo Urías-Sotomayor [1], Guillermo Rodríguez-Domínguez [2], José Adán Félix-Ortiz [2], Gilberto G. Ortega-Lizárraga [3], Horacio A. Muñoz-Rubí [3] and Eugenio Alberto Aragón-Noriega [1,*]

[1]  Unidad Guaymas del Centro de Investigaciones Biológicas del Noroeste, S.C. Km 2.35 Camino a El Tular, Estero de Bacochibampo, Guaymas 85454, Sonora, Mexico; rsotomayor@pg.cibnor.mx

[2]  Facultad de Ciencias del Mar, Universidad Autónoma de Sinaloa, Paseo Claussen S/N, Mazatlan 82000, Sinaloa, Mexico; guirodom@uas.edu.mx (G.R.-D.); feocabo@uas.edu.mx (J.A.F.-O.)

[3]  Centro Regional de Investigación Acuícola y Pesquera Mazatlán, Calzada Sábalo-Cerritos s/n, Contiguo Estero del Yugo, Mazatlan 82112, Sinaloa, Mexico; gilberto.ortega@inapesca.gob.mx (G.G.O.-L.); horacio.munoz@inapesca.gob.mx (H.A.M.-R.)

*  Correspondence: aaragon04@cibnor.mx; Tel.: +52-622-221-2238

**Abstract:** A stock reduction analysis (SRA) of bigeye croaker *Micropogonias megalops* was performed based on commercial catch data. SRA solutions were restricted to a 2011 bigeye croaker stock biomass estimate of 14,412 t. The viable solution indicated a reduction in stock of 73.6% from 1983 to 2020 with an initial biomass of 22,186 t. In addition, a possible effect of hyperstability of the stock was evaluated by applying different versions of the Cobb–Douglas catch function. The most probable function based on a multi-model selection procedure was the one wherein the catch does not depend on biomass and is directly proportional to the applied fishing effort of small boats (~7 m) and vessels (~24 m). This situation suggests that in a free access regime, fishing can deplete the resource until it collapses, without observing a significant reduction in its catches until the event is very close.

**Keywords:** bigeye croaker; Cobb–Douglas; stock reduction analysis; upper Gulf of California

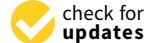

## 1. Introduction

The bigeye croaker *Micropogonias megalops* is an endemic fish to the Gulf of California that has been subject to commercial fishing since 1991, by artisanal fishing as well as by vessels with trawling equipment that operate in the upper Gulf of California. This species, together with the blue shrimp *Penaeus stylirostris*, the gulf corvina *Cynoscion othonopterus* and the Spanish mackerel *Scomberomorus sierra* (also known as Pacific sierra) constitutes one of the four most important fishery resources of artisanal fisheries in the upper Gulf of California [1,2]. The catch of bigeye croaker takes place mainly between April and June each year [3] when it aggregates to reproduce. The catches are made with gillnets on artisanal fishing small boats and with trawl nets on larger shrimp fishing vessels (Figure 1).

Most of the catches are made within the protected natural area called Biosphere Reserve of Upper Gulf of California and Colorado River Delta (UGC) [4] that was delimited since 1993, which covers an area of 934,756 ha, mainly to preserve natural environments and the most fragile ecosystems, their diversity of flora and fauna species, particularly endemic, threatened and endangered, such as vaquita *Phocoena sinus* and totoaba *Totoaba macdonaldi*—the latter is illegally captured due to its high economic value on the illicit market.

The biological information of *M. megalops* is limited. There is scarce information about growth and natural mortality and only one estimate of biomass from 2011 [5,6]. There are no studies on the dynamics of the resources that allow to know the status of the stock. Measuring the stock status is difficult because the fishing is carried out on reproductive aggregations, so the catch does not indicate the density of the population. There are no specific fishing regulations for this species other than those applicable to other marine fin-fish

species in general and the restrictions imposed by the National Commission of Protected Natural Areas. Starting in 2015, then in 2017 and 2018 [7,8], the temporary suspension of the use of gillnets gear was established in the polygon delimited for the protection of the vaquita marina and totoaba, presenting a reduction in the catches of *M. megalops* and conflict between authorities and fishers over these environmental provisions. However, in 2019, fishing was allowed, and there was a rebound in commercial catches. However, definitively in 2020 [9], a zone of prohibition of catching with gillnets was declared in the UGC, which will possibly impact the fishery.

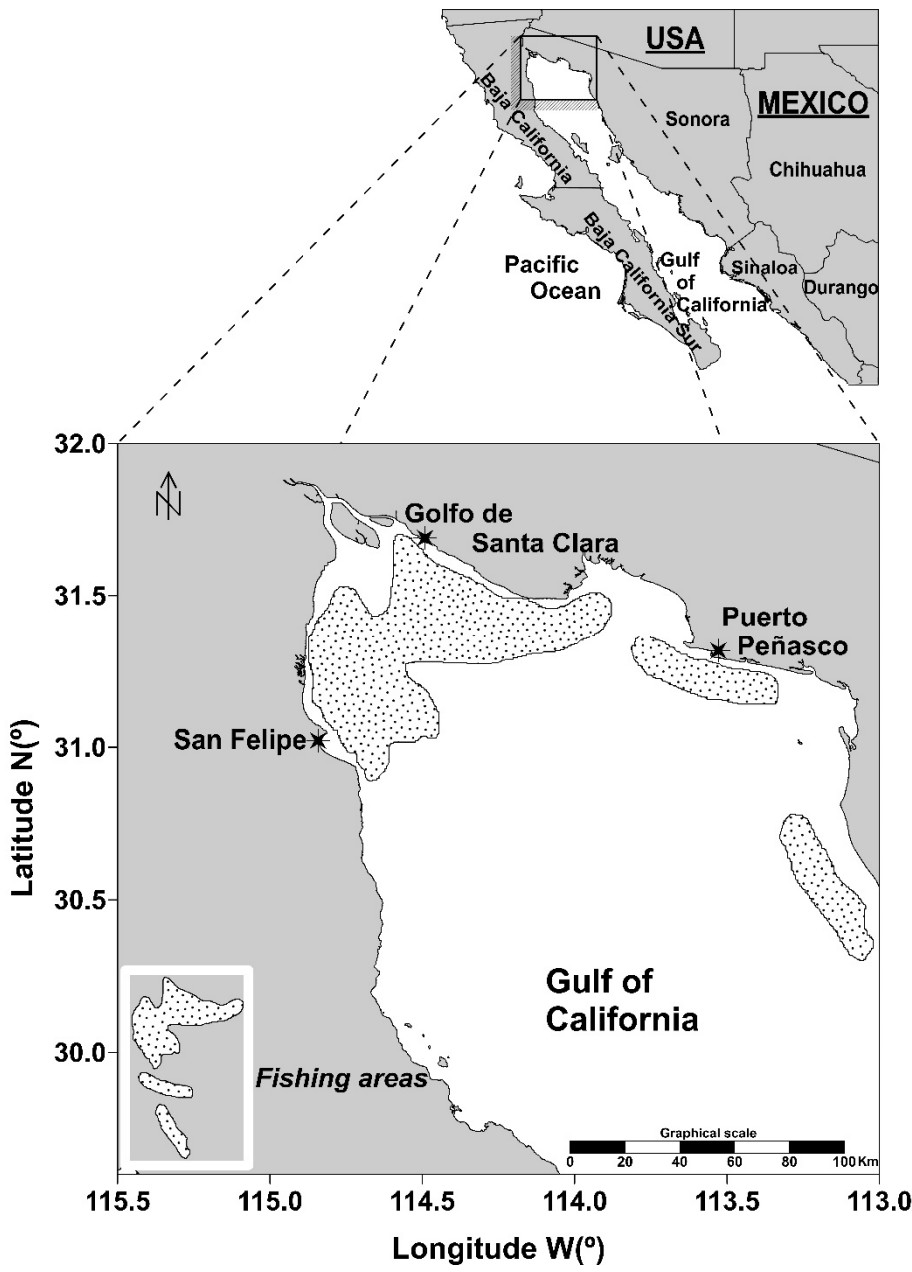

**Figure 1.** Catches areas of bigeye croaker *Micropogonias megalops* at upper Gulf of California.

Due to the importance that this fishery represents in the artisanal fishing of the UGC, it is necessary to have a stock assessment on which to support management procedures. This work provides an evaluation of the bigeye croaker stock based on a stock reduction analysis (SRA) suitable for resources with limited information and using data from commercial catches and effort, as well as an evaluation of the possible effect of hyperstability of the stock. It is

worth noting that hyperstability occurs when catch rates remain high even as fish population decline. The best-known case is in schools of pelagic, but in the case of fish reproductive aggregations, this phenomenon is also possible, even in pelagic or demersal stock.

## 2. Materials and Methods

The information on the recorded catches of *M. megalops* in the upper Gulf of California, Mexico from 1991 to 1999 for the artisanal and industrial fleets comes from reference [3] and from 2000 to 2020 from the Comisión Nacional de Acuacultura y Pesca. This information is registered in the fishing and aquaculture computer system (provided in Excel format), whose records are supported by the arrival notices, through which, in a way mandatorily, fishing permit holders must declare their catches [10] (Figure 2). The fishery rebound started after 2010 was analyzed [1,2], and they identify four production periods that resumed: low production, fleet expansion, overfishing and the standardization of catch yields.

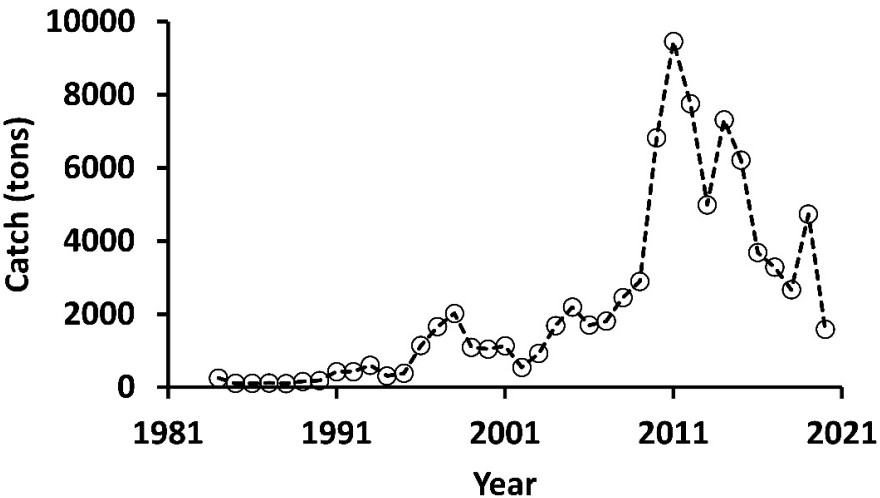

**Figure 2.** Annual catches of bigeye croaker *Micropogonias megalops* in the upper Gulf of California.

Bigeye croaker catch data from 1984 to 2020 were treated through the stock reduction analysis procedure [11]. Given a series of annual catch data, the stock reduction analysis consists of obtaining the solution of the Baranov's catch equation [12], which links the size of the stock, the instantaneous rate of natural mortality, the fishing mortality for each year and the size of the annual catch. Additionally, the change in the initial biomass due to the years of fishing is simultaneously resolved as a reduction ratio *Bfinal/Binitial*. The biomass at the beginning of each year results from the decrease in the biomass at the beginning of the previous year because of the total mortality (natural and due to fishing) plus the biomass of the recruitment (it includes growth and recruitment).

The three basic equations are:

$$C_i = \frac{B_i F_i \left(1 - e^{-F_i - M}\right)}{(F_i + M)} + R, \tag{1}$$

$$B_i = B_{i-1} e^{-F_i - M} + R, \tag{2}$$

$$P = \frac{B_1}{B_{n+1}}, \tag{3}$$

where $C_i$, $B_i$, and $F_i$ are the estimated catch, biomass, and instantaneous fishing mortality rate in year $i$, respectively. $M$ is the natural mortality, $R$ is the recruitment, $P$ is the ratio of reduction in the stock in the years of fishing, and $n$ is the number of years of catch analyzed.

The $n + 1$ nonlinear equations of catch and the reduction ratio ($P$) are solved simultaneously by interactively looking for the parameters $B_1$, $M$, $F_i \ldots F_n$, $P$, and $R$. Setting $B_1$, $M$, and $P$, the equations are adjusted to find $F_i \ldots F_n$, and $R$.

Considering a biomass at the beginning of 2011 of 14,412 t [6] and given that 2011 was when the maximum catch of the series was recorded, it was considered that in that year the reduction in the stock could be between 0.4 and 0.7 with respect to the virgin biomass, for which the initial biomass interval ($B_1$) to explore the solution to the catch equations and stock reduction rate was empirically defined between 20,000 and 35,000 t, and the natural mortality was set at $M = 0.51$ (C.I. 95% 0.41, 0.60) [6]. $M$ values were determined from six empirical equations [6]. Since 2011, the biomass estimate was made in the fish reproductive aggregation areas of croaker. As with the catch data, it only includes the adult population. As the beginning of the catch series can be assigned to a time when the population was virgin or relatively unexploited, the expected recruitment line was used. In other words, at the beginning of the time series, the population is at equilibrium; thus, recruitment must offset death [11]. Then, found that $P$ and $B_1$ that are consistent with the replacement concept because they intersect with the expected recruitment line. Subsequently, $M$ and these values of $P$ and $B_1$ are fixed and explored for the values of $F_i$ and $R$, which solve the Baranov catch equations for each year of the catch series. The acceptable solution of this set of equations by the least squares procedure, according to [11], means that the sum of the squared error (SSQ), of comparing the observed and estimated catch, is almost zero. For this work, SSQ was estimated, and those combinations of $P$ and $B_1$ that result in an evidently high SSQ were discarded.

There is no single solution to this set of equations, so additional information is required to find the solution. In this work, information on the estimated biomass of bigeye croaker stock in 2011 was used (14,412, C.I. 95% 1868, 21,615 t) (6). SRA is a deterministic procedure; however, using the 95% confidence interval of $M$ and biomass in 2011 allows to approximate the uncertainty estimate of the stock condition.

To explore the effect of unreported catch in the official record, the original catch was increased by 20% and 40% and then the SRA was applied assuming $M = 0.51$ and restricting the solution to a $B_{2011} = 14,412$.

To explore an effect of hyperstability in the bigeye croaker stock, a Cobb–Douglas catch function [13] was adjusted with the estimated biomass data, the effort of both fleets and total catch in the period from 2008 to 2020 with the following equation:

$$C_e = A_{sb} f_{sb}^{v1} B^{w1} + A_{sh} f_{sh}^{v2} B^{w2}, \tag{4}$$

where $Ce$ is the estimated total catch; $A_{sb}$ and $A_{sh}$ are constants associated with small boats and vessels, respectively; $f_{sb}$ and $f_{sh}$ are the effort of small boats and vessels, respectively; $B$ is the biomass of the stock; and $v_1$, $v_2$, $w_1$, and $w_2$ are parameters associated with the effort and biomass of the stock. The values of the parameters $v$ and $w$ will allow to detect externalities of the effort and the linearity or not of the population density with the size of the stock. Values of $v$ and $w$ equal to 1 imply a constant catchability, so the catch depends proportionally on the size of the fleet and size of the stock, while values of $v < 1$ imply externalities of effort congestion, while $v > 1$ implies everything opposite. The value of $w < 1$ implies that the population density does not decrease linearly with the size of the stock, as in the case of hyperstability, and $w = 0$ implies that the catch does not depend on the size of the stock.

The parameters $A$, $v$, and $w$ of both fleets were adjusted for the maximum loglikelihood ($LL$), assuming a lognormal error distribution with the following $LL$ function:

$$LL = \left(\frac{-n}{2}\right)[\ln(2\pi) + 2\ln(\sigma) + 1] - \sum \ln(C_o), \tag{5}$$

$$\sigma = \sqrt{\frac{\sum \ln(C_o/C_e)^2}{n}}, \tag{6}$$

where $Co$ is the observed catch.

Several versions of the catch function were compared, fixing some parameters to 0 or 1 as in Table 1.

**Table 1.** Values for parameters according to the catch function assessed.

| Parameter | Version of Catch Function | | | |
|---|---|---|---|---|
| | **1** | **2** | **3** | **4** |
| $v_1$ | * | 1 | * | 1 |
| $w_1$ | * | 1 | 0 | 0 |
| $A_{sb}$ | * | * | * | * |
| $V_2$ | * | 1 | * | 1 |
| $w_2$ | * | 1 | 0 | 0 |
| $A_{sh}$ | * | * | * | * |
| $\phi$ | 7 | 3 | 5 | 3 |

* Freely estimated. The numbers are the restrictions applied to the catch function and $\phi$ is the number of parameters involved in the fit ($\sigma$ is included).

In the first version, all the parameters are obtained without restriction; in the second version, the restrictions mean that the catch is linearly proportional to the effort and biomass in both fleets. The third version means that the catch depends on the effort with some externality, but it is independent of the size of the stock, and in the fourth version, the catch depends linearly on the effort, without externalities, but independent of the size of the stock.

## 3. Results

Table 2 shows the SSQ and biomass estimated at the beginning of 2011 of bigeye croaker that results from solving the Baranov capture equations in the SRA, for three values of $M$ and combinations of $B_1$ and $P$ that are consistent with the concept replacement because they intersect the expected recruitment line. It is observed that with $M$ of 0.41 and 0.51 and $B_1$ below 20,000 t, they do not result in a good solution to the catch equations because the SSQ rises considerably. The same happens but with $B_1$ less than 15,000 t when $M = 0.6$.

**Table 2.** Acceptable solutions of SRA of bigeye croaker *M. megalops* for three *M* values and information on biomass in early 2011. Bold numbers are approximations to media and upper limit of C.I. 95% of $B_{2011}$.

| *M* | *P* | $B_1$ | $B_{2011}$ | *R* | SSQ |
|---|---|---|---|---|---|
| | 0.5 | 15,527 | 8905 | 5796 | 10,174,699 |
| 0.41 | 0.62 | 20,701 | 11,766 | 6955 | 13,314 |
| | 0.6639 | 23,339 | **14,414** | 7886 | 0.0000 |
| | 0.74 | 30,665 | **21,658** | 10,321 | 0.0000 |
| | 0.6 | 14,592 | 8679 | 6293 | 5,960,131 |
| 0.51 | 0.7 | 19,413 | 11,654 | 7752 | 0.0000 |
| | 0.7359 | 22,191 | **14,416** | 8872 | 0.0000 |
| | 0.8 | 29,603 | **21,770** | 11,832 | 0.0000 |
| | 0.6 | 11,508 | 7623 | 6000 | 16,104,726 |
| 0.6 | 0.7 | 15,435 | 9146 | 7125 | 790,845 |
| | 0.786 | 21,361 | **14,411** | 9641 | 0.0000 |
| | 0.835 | 28,641 | **21,630** | 12,927 | 0.0000 |

The SRA solutions that replicate the estimated average biomass (14,412 t) in 2011 (6) were obtained with $B_1 = 23,339$ t, $P = 0.6639$ and $M = 0.41$, but also with $B_1 = 21,361$ t, $P = 0.786$ and $M = 0.6$. The upper limit of the C.I. 95% of the biomass estimated in 2011 (21,615 t) was replicated in the SRA with $B_1 = 30,665$ t, $P = 0.74$ and $M = 0.41$, but also with $B_1 = 28,641$ t, $P = 0.835$ and $M = 0.6$. The lower limit of the C.I. 95% of $B_{2011}$ (1868 t) could not be replicated in the SRA because there was no acceptable solution for any of the $M$, $P$, and $B_1$ values. The minimum acceptable solution was obtained with $B_1 = 15,435$ t, $P = 0.7$, and $M = 0.6$, resulting in a $B_{2011}$ of 7125 t. Considering the acceptable solutions of

SRA for the C.I. 95% for $M$ and $B_{2011}$, it can be accepted that the C.I. 95% for $B_1$ are: 22,191 (15,435–30,665) t, $P$ = 0.7359 (0.62–0.835), and $R$ = 8872 (6,955–12,927) t.

The effect of a constant unreported catch rate of 20% and 40% was a 3% and 6% reduction in the rate of decline of biomass, respectively (Table 3). On the other hand, the biomass increased by 7% and 11% and recollecting also increased by 6% and 12%. However, it is fair to mention that in the 40% catch compensation scenario, the acceptable solution was at the limit. A higher rate of underreporting is not an acceptable solution if $B_{2011}$ = 4412 t.

**Table 3.** SRA results under different unreported catch rate scenarios assuming $M$ = 0.51 and $B_{2011}$ = 14,412 t.

| Setting (%) | $P$ | $B_1$ | $R$ | $B_{2011}$ |
|---|---|---|---|---|
| 0 | 0.74 | 22,186 | 8869 | 14,410 |
| 20 | 0.71 | 23,729 | 9478 | 14,418 |
| 40 | 0.67 | 24,797 | 10.039 | 14,418 |

The estimated biomass of bigeye croaker after SRA shows an extremely similar performance for the curves with different values of $M$ (Figure 3A). The greatest variation occurs when the acceptable solution is restricted with the information of the biomass of the stock in 2011 (and its C.I. 95%) previously estimated. However, in general, there is a drastic decrease in the size of the stock after 2010, and a rapid recovery after 2016 that continues until the beginning of 2011. Given the different SRA solutions for different values of $M$ and biomass in 2011, one way to measure the relative reduction of the stock is through the $B_{actual}/B_1$ ratio. By early 2021, the stock of bigeye croaker in the upper Gulf of California is above $0.5B_1$ (Figure 3B), and considering that $B_1$ approximates the size of the virgin population, the stock is in good condition.

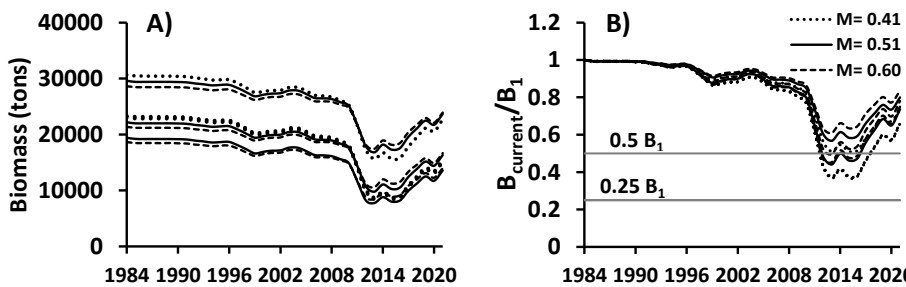

**Figure 3.** Biomass from stock reduction analysis solution for *Micropogonias megalops* in the upper Gulf of California. For each type of line, the upper line is the SRA solution restricted by $B_{2011}$= 21,361 t, the middle line is the SRA solution restricted by $B_{2011}$= 14,412 t, and the lower line is the SRA minimum acceptable solution. (**A**) estimated biomass; (**B**) estimated biomass ratio of current versus $B_1$.

The best production function was that of case 4 with three parameters to estimate. It obtained the lowest value according to the Akaike Information Criterion (AIC) and an Akaike weight of 0.59. According to this function, the catch of both fleets does not depend on the biomass of the stock and is directly proportional to the effort of each fleet (Table 4).

**Table 4.** Parameters of catch function fit and data for multi-model selection.

| Case | Parameter | $v_1$ | $w_1$ | $A_{sb}$ | $v_2$ | $w_2$ | $A_{sh}$ | AIC | $W$ |
|---|---|---|---|---|---|---|---|---|---|
| 1 | 7 | 3.18 | 0.53 | 0.00 | 1.25 | 0.42 | 12.37 | 335.38 | 0.00 |
| 2 | 3 | **1.00** | **1.00** | 0.00 | **1.00** | **1.00** | 0.00 | 237.45 | 0.01 |
| 3 | 5 | 2.21 | **0.00** | 0.00 | 1.34 | **0.00** | 7.27 | 230.13 | 0.39 |
| 4 | 3 | **1.00** | **0.00** | 7.82 | **1.00** | **0.00** | 23.42 | 229.31 | 0.59 |

Bold numbers are fixed restrictions. $W$ is the Akaike weight (percent of evidence in favor a model).

## 4. Discussion

*Micropogonias megalops* is one of the four most important species of commercial interest of artisanal fisheries in the upper Gulf of California [14,15]. However, biological studies aimed at evaluating the stock of this species is limited. Most have focused on growth [5,6], and the natural mortality and biomass of this species have only once been estimated [6]. In this context, *M. megalops* is a species little studied in the region in terms of fishery biology, so its fishery can be considered as a data-poor fishery, which limits its analysis and management, with implications for its exploitation. The present work constitutes the first attempt to evaluate the dynamics of this species, addressing the stock reduction analysis as a procedure for fisheries with limited information. This procedure has multiple solutions, so the additional information is important to narrow down the most likely option. In this case, the assumption of the initial biomass limits between 20,000 and 35,000 t was a good choice since mayor solutions were within this interval and the hypothesis of an initial biomass outside this interval does not converge on the biomass estimated in 2011.

As the catch series begins in the first years of commercial catch of this species, the solution for $B_1$ must be very close to the carrying capacity ($k$), and in 2011, when the highest commercial catch was obtained, the reduction in biomass was between 0.6 and 0.76. For the year 2012, with relatively high catches, the biomass was reduced to around 0.43 to 0.60, remaining so until 2015. It is notable that after the restriction to fishing in the upper Gulf of California in 2015, 2017, and 2018, there is a stock recovery, up to 0.60 (0.6 to 0.8) in early 2019.

The historical series of available bigeye croaker catches begins in 1984 at an early stage of the fishery, so it can be assumed that the initial biomass is that of the virgin biomass. This situation allows us to use the expected recruitment line (6) to narrow down the acceptable solution of the bigeye croaker SRA from the upper Gulf of California, from the widest number of acceptable solutions possible. Moreover, having an estimate of biomass by the swept area method with its confidence intervals allowed a better definition of the acceptable solution of the Baranov system of capture equations used in the SRA. In addition, a mean estimate and confidence intervals of natural mortality were available, which made it possible to have a wide range of solutions which can be associated with means and confidence intervals of the SRA output parameters. SRA solutions were the most similar between different values of $M$ (C.I. 95%) than between biomass (C.I. 95%) values used as restrictors. This is largely due to the fact that the confidence interval estimated for $B_{2011}$ by the swept area method was very broad given the nature of the aggregate distribution of the bigeye croaker. A $B_{2011}$ C.I. 95% that varies from 2885 to 21,655 t and is used to limit the extreme solutions of SRA influenced a wide range of the output results; however, with the lower limit of this biomass, it was not possible to find an acceptable solution, which assumes that although, statistically, the swept area method defines it as a lower limit, in reality, the size of the stock was never able to reach this point. The mean and upper limit of the estimated $B_{2011}$ allowed to limit an acceptable solution of the SRA for the three levels of $M$; on the other hand, it was not possible to find an acceptable solution putting as a restriction the lower limit of the $B_{2011}$ for any of the three values of M However, using the SRQ as a guide, it was possible, empirically, to obtain a solution with a minimum acceptable $B_{2011}$ estimate for each level of M. An acceptable SRA estimate of $B_{2011}$ was around 7000 t, far from the estimated 2885 t with the swept area method. Thus, the reduction in the stock at the beginning of 2021 must have been in the range of $P = 0.62$ to 0.835 of the virgin biomass, which was the initial biomass in 1984. From this, it is concluded that the current condition of the stock of bigeye croaker in the upper Gulf of California is good, because it is generally acceptable to keep biomass above $0.5Bv$. Note that when the catch series starts, from the first years of the fishery, the biomass was virgin, or with a low exploitation, the replacement line was used and then $B_1$ became $Bv$. The SRA is a simple and effective tool to estimate the condition of the stock in fishery resources with little scientific information, but it depends on a historical series of reliable catches. In Mexico, it is presumed that the catch statistics are a lower percentage than what is actually obtained. Fishers usually register a lower percentage than what they actually catch. The explanation of this behavior

is outside the scope of this work. The exploration of unrecorded catch scenarios indicates that the effect translates into an overestimation of the rate of decline of biomass ($P$) and an underestimation of the initial biomass and recruitment. However, the biomass reduction rate, which indicates the condition of the stock, was reduced by just 0.07 if an extreme unreported catch of 40% is assumed. Considering the lowest value of $P = 0.62$ estimated with the original data, assuming an extreme of 40% of unrecorded catch, when correcting the catch, the SRA would yield a $P = 0.56$ and would not change the conclusion of the condition of the stock. However, it could also be seen that by compensating a 40% of unreported catch, the minimum acceptable solution was at the limit, suggesting that the underreporting in this fishery should be less than 40%. Another accepted weakness of SRA (6) is that it considers recruitment constant, and it is known that in many marine fish populations there is a great interannual variability of recruitment. However, it must also be seen that the $R$ term of the SRA proposed by (6) includes biomass recruitment and growth, so that if there is a density-dependent relationship between recruitment and biomass growth, both contributions can complement each other and maintain an $R$ (such as that defined in [6]) constant. An attempt to apply the generalized SRA [16] that considers variable recruitment was not successful in finding acceptable solutions at any level of $M$.

The two best catch models agree that the catch of *M. megalops* is independent of the size of the stock; therefore, the catch per unit of effort (CPUE) is also not an indicator of the abundance of the stock. For this reason, surplus production models such as the Schaefer model [17], which is commonly applied in fisheries with limited information, are not adequate to evaluate this resource. In a situation such as this, wherein the catch does not depend on the size of the stock, there is never an equilibrium; in a situation of free access, the effort could reduce the resource, without observing a significant decrease in the catch, up to a very near collapse. Fortunately, this situation has not occurred, nor have the stock levels reached a limit point, such as $0.25B_1$, in which recruitment was put at risk.

Apparently, there is not much interest in the regulation of this resource in the National Fisheries Charter of Mexico, which is binding in making management decisions. There is also no technical file for bigeye croaker *M. megalops*; it only appears as an associated species to fishing for gulf corvina *C. othonopterus*. The effort regulations applied to this fishery have been due to environmental provisions for the protection of the porpoise *P. sinus* and the totoaba *T. macdonaldi* and/or the price of the product. Although the bigeye croaker is one of the four most important species of artisanal fisheries in terms of effort, catch, and income in the upper Gulf of California [4], the bigeye croaker occupies the second place of catch, but fourth place in revenue. It seems that the price of this product limits the extent of its catches. Most fishers agree that income from fishing for bigeye croaker is very low. Erisman et al. [4] pointed out that in the UGC, the income from fishing for gulf corvina *C. othonopterus* is 2.5 times higher than that from catching bigeye croaker *M. megalops*, 1.3 higher with the Pacific sierra *S. sierra* and up to 6.2 times higher with blue shrimp *P. stylirsotris*. From 2015 to 2018, and finally in 2020, the prohibition of gill gear such as those used in catching bigeye croaker in the Upper Gulf of California and the Colorado River Delta Biosphere Reserve was established, which will undoubtedly limit fishing efforts. This situation can improve the condition of the stock. However, it is necessary to continue to develop more information that allows an assessment of the stock with better information, such as that of the catch at age.

## 5. Conclusions

The most probable function based on a multi-model selection procedure was the one wherein the catch does not depend on biomass and is directly proportional to the applied effort of small boats (~7 m) and vessels (~24 m). This situation suggests that in a free access regime, fishing can deplete the resource until it collapses, without observing a significant reduction in its catches until the event is very close. According to the results, the bigeye croaker stock in the upper Gulf of California keeps to an underexploited level. Irrespective of the previous conclusion and considering that the species' reproductive behavior is to

travel long distances to spawn in aggregations that persist for approximately three months, the fishery must be managed with extreme caution to keep it suitable.

**Author Contributions:** Conceptualization, R.U.-S.; G.R.-D. and E.A.A.-N.; methodology, R.U.-S.; and G.R.-D.; formal analysis, R.U.-S.; G.R.-D. and E.A.A.-N.; investigation, R.U.-S. and E.A.A.-N.; resources, R.U.-S.; G.R.-D. and E.A.A.-N.; data curation, R.U.-S.; G.R.-D. and E.A.A.-N.; writing—original draft preparation, R.U.-S.; G.R.-D.; J.A.F.-O.; G.G.O.-L.; H.A.M.-R. and E.A.A.-N.; writing—review and editing, R.U.-S.; G.R.-D.; J.A.F.-O.; G.G.O.-L.; H.A.M.-R. and E.A.A.-N.; supervision, E.A.A.-N. All authors have read and agreed to the published version of the manuscript.

**Funding:** This research received no external funding.

**Acknowledgments:** The first author (CVU 634251) is grateful for the support provided by the Consejo Nacional de Ciencia y Tecnología (National Council of Science and Technology), through the postdoctoral stay program in Mexico at Centro de Investigaciones Biológicas del Noroeste, S. C. We also thank Edgar Alcántara-Razo, from CIBNOR-Guaymas Applied Ecology and Fisheries Lab for improving the figures.

**Conflicts of Interest:** The authors declare no conflict of interest.

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
