# Peer review of "Stock Reduction Analysis of Bigeye Croaker Micropogonias megalops in the Upper Gulf of California, Mexico"

_fishes, doi:10.3390/fishes7010015_

Round 1

Reviewer 1 Report

“Stock reduction analysis of bigeye croaker Micropogonias megalops in the upper Gulf of California, Mexico”, Ref N°: fishes-1512018 by Ricardo Urías-Sotomayor et al. After reading the article, I notice that:

(1) The present version of the manuscript seems to me more elaborate and better organised and presented. The state of knowledge on the species and the topic have been improved.

(2) The study area is be accompanied by a map indicating the location of data collection. More information on M. megalops catches have been added.

(3) The results and discussion seem to be improved.

I thank the authors to have considered all the questions/comments/suggestions raised during the first evaluation process of this manuscript. Their adaptations and explanations are sufficient to allow the decision to proceed to a minor revision of the manuscript.

I must congratulate the authors for the effort put into the revision of the manuscript and its resubmission. You have addressed all the aspects suggested. I thank you.

I have only minor suggestions or comments:

L66-72 I suggest here that the authors comment on Figure 2 by explaining the origin of the peaks observed after 2010.

L161 “similar picture”, I suggest using a more appropriate term. It needs to be reworded.

L262 “like the [16]”, please highlight the author. This style seems inappropriate. I suggest "like the Schaefer [16]" or "like the Schaefer model [16]". See also L279 “[4] point out”.

L289-293 Here authors should provide, based on their findings, at worldwide level as well as in context of the Gulf of California some specific recommendations/implications for conservation and preservation as well as for the sustainability of the bigeye croaker Micropogonias megalops in the future.

Author Response

(1) The present version of the manuscript seems to me more elaborate and better organized and presented. The state of knowledge on the species and the topic have been improved.

Thank you so much for this comment

(2) The study area is be accompanied by a map indicating the location of data collection. More information on M. megalops catches have been added.

Thank you so much for this comment

(3) The results and discussion seem to be improved.

Thank you so much for this comment

I thank the authors to have considered all the questions/comments/suggestions raised during the first evaluation process of this manuscript. Their adaptations and explanations are sufficient to allow the decision to proceed to a minor revision of the manuscript.

Thank you so much for this comment. We made all minor corrections suggested

I must congratulate the authors for the effort put into the revision of the manuscript and its resubmission. You have addressed all the aspects suggested. I thank you.

Thank you so much for this comment

I have only minor suggestions or comments:

L66-72 I suggest here that the authors comment on Figure 2 by explaining the origin of the peaks observed after 2010.

This text was added (lines 76-78 new version): The fishery rebound started after 2010 was analyzed [1,2] and they identify four production periods that resumed as low production, fleet expansion, overfishing and standardization of catch yield

L161 “similar picture”, I suggest using a more appropriate term. It needs to be reworded.

The new text in lines 186-187 is: SRA shows an extremely similar performance for the curves with different values of M

L262 “like the [16]”, please highlight the author. This style seems inappropriate. I suggest "like the Schaefer [16]" or "like the Schaefer model [16]". See also L279 “[4] point out”.

Done: see the new lines 305-306 and 322

L289-293 Here authors should provide, based on their findings, at worldwide level as well as in context of the Gulf of California some specific recommendations/implications for conservation and preservation as well as for the sustainability of the bigeye croaker Micropogonias megalops in the future.

This text was added (lines 336-341 new version): According to the results the bigeye croaker stock in the upper Gulf of California keeps underexploited level. Irrespectively of the previous conclusion and considering that the species reproductive behavior is to travel long distances to spawn in aggregations, that persist for approximately three months, the fishery must be managed with extremely caution to keep it suitable.

Reviewer 2 Report

This is a good manuscript to study the stock reduction analysis of bigeye croaker Micropogonias megalops in the upper Gulf of California, Mexico, I would recommend publication after revisions.

1, The caption of Figure 3 needs to be re-written. Figure 3A-3C may be combined into one figure, similar to Figure 3D.

2, Bold numbers are not explained in Table 2.

3, “W” is not explained in Table 3.

4, There are recent published software to do the data limited analysis including SRA, such as DLMtool (https://www.datalimitedtoolkit.org/), it may be interesting to see a comparison.

5, Usually hyperstability is for the pelagic schooling fish such as anchovy, in this study the species of croaker is demersal.

Author Response

This is a good manuscript to study the stock reduction analysis of bigeye croaker Micropogonias megalops in the upper Gulf of California, Mexico, I would recommend publication after revisions.

Thank you so much for this comment. We made all minor corrections suggested

1, The caption of Figure 3 needs to be re-written. Figure 3A-3C may be combined into one figure, similar to Figure 3D.

Done. See the new figure 3

2, Bold numbers are not explained in Table 2.

Done: see the heading of table 2 in lines 174-175

3, “W” is not explained in Table 3.

Done. See the foot of previous table 3 in line 210. In new version is table 4

4, There are recent published software to do the data limited analysis including SRA, such as DLMtool (https://www.datalimitedtoolkit.org/), it may be interesting to see a comparison.

We are exploring the suggested software, but we believe it we be difficult to include the comparison in this paper. Surly we used it in future analysis

5, Usually hyperstability is for the pelagic schooling fish such as anchovy, in this study the species of croaker is demersal.

This text was added in lines 65-68: It is worth noting that hyperstability occur when catch rates remain high even as fish population decline. The best-known case is in schools of pelagic, but in the case of fish reproductive aggregations this phenomenon is also possible. even if they are pelagic or demersal stock.

Reviewer 3 Report

Overview:

This paper provides a population dynamic analysis, specifically a stock reduction analysis (SRA), for the bigeye croaker, a fish commonly caught by commercial fisheries, but with a relatively low economic value, thus is not a major target. It lacks basic biological and fisheries data, but it does have one estimate of biomass. The authors use a sensitivity approach to apply an uncertain catch history to uncertain inputs in order to produce an estimate of stock status. While other methods could be used, SRA is one of suitable methods, and I think the authors did a good job on not focusing on one set of results, but rather scanning across a suite of scenarios to indicate stock status. In this case, the results of a stock status above the target reference point of B50% is robust to major sources of uncertainty. The authors also consider whether the use of catches and effort data may be subject to hyperstability (due to the aggregating behavior of this species), which they find it is, and thus recommend caution when using methods that rely on a better relationship between catch per unit effort and abundance. Overall I think the authors have done a good job with limited information. My biggest suggestions come from adding more details about the assumptions behind the method, and elevating some of the results presented only in the Discussion (that regarding different assumptions about the catch history) to the main Methods and Results section.

Major considerations

  • The assumptions of this approach are key to the implementation. There are several places more details are needed as to where values were produced:
    • There is a mention of a biomass estimate in 2011. Where did this come from? Is it an absolute biomass of the whole population, just the spawning population, or something else?
    • Related to the above question, there is an implicit assumption of what biomass is (and defined through the catch selectivity curve), but it is not made explicit. Please given some detail as to what the B value includes, as I am not sure it is total biomass.
    • The stock reduction in 2011 is based on what? Is it expert opinion?
    • Instead of just citing a reference, maybe say a bit more where the natural mortality value comes from. Is it directly measured or empirically derived?
    • Are catches being used total removals from all fisheries? Is there any dead bycatch not being considered? Anyone of these things can mess us the estimation. Lines 235-237 provide some insight here, and provides what seems to be additional results. Why are those not in the main Methods and Results section? Those results are as important as any presented. I highly recommend incorporating the exploration of uncertainty in catches into the main Methods and Results sections, not just in the Discussion.

Minor considerations

  • Lines 97-99: More description of what you mean by “the expected recruitment line was used” is needed. I think you mean to say that at the beginning of the time series the population is at equilibrium, thus recruitment must offset death. More detail will help the reader understand the many assumptions inherent in an SRA.
  • The Conclusions section is odd. It just presents conclusions from the hyperstability analysis. As this was just a small part of the overall paper, why is it the only thing in the Conclusions section?

Edits/Suggestions:

  • Lines 36-42: This is a run on sentence. Please break this up into 2-3 sentences.
  • Lines 43-44: “There is scarce information about growth and mortality, and only one estimate of biomass from 2011.”
  • Line 45: “… knowing the status of the stock. Measuring stock status is difficult because…”
  • Line 68: ¨Pesca,,- why the quotes and double commas? You could also end the sentence here and start the next as “This information is registered…”
  • Line 186, 272: “species”
  • Line 208: Should “But” be capitalized? It follows a period. I don’t think a comma should be used, as it would create a run-on sentence, so breaking the sentences apart makes sense.
  • Line 210: “acceptable solution of the acceptable resolution…”—I am not sure what the “acceptable resolution” means. Is this the units of biomass, a time period, something else? It refers to the Baranov catch equation, but it is not clear what it means.
  • Line 213: What is “RAS”—should this be SRA?
  • Lines 214-216: How does greater variation come from restricting confidence intervals? I suspect this is a grammatical error that needs to be fixed.
  • Line 233: Should Bv be B1?
  • Line 233: I do not think it can be said that SRA is reliable. It is what it is, and under many assumptions it can offer some insight into the population dynamics. But to say it is reliable oversells it, as what it is reliable for is undefined here. It happens that the stock status seems above the target even under a variety of scenarios, so it can be said the stock status results are robust to explored uncertainty.
  • Line 262: “Like the…” what? It just gives a reference instead of a name of the method.

Author Response

This paper provides a population dynamic analysis, specifically a stock reduction analysis (SRA), for the bigeye croaker, a fish commonly caught by commercial fisheries, but with a relatively low economic value, thus is not a major target. It lacks basic biological and fisheries data, but it does have one estimate of biomass. The authors use a sensitivity approach to apply an uncertain catch history to uncertain inputs in order to produce an estimate of stock status. While other methods could be used, SRA is one of suitable methods, and I think the authors did a good job on not focusing on one set of results, but rather scanning across a suite of scenarios to indicate stock status. In this case, the results of a stock status above the target reference point of B50% is robust to major sources of uncertainty. The authors also consider whether the use of catches and effort data may be subject to hyperstability (due to the aggregating behavior of this species), which they find it is, and thus recommend caution when using methods that rely on a better relationship between catch per unit effort and abundance. Overall, I think the authors have done a good job with limited information. My biggest suggestions come from adding more details about the assumptions behind the method and elevating some of the results presented only in the Discussion (that regarding different assumptions about the catch history) to the main Methods and Results section.

Thank you so much for this comment. We made all corrections suggested. Responding to the last sentence of this point we add a text in method (lines 123-125) and results (lines 177-185) sections to clarify the unrecorded catch that in previous version was only in the discussion section. Also, we change a paragraph in discussion section (lines 270-294).

Major considerations

The assumptions of this approach are key to the implementation. There are several places more details are needed as to where values were produced:

There is a mention of a biomass estimate in 2011. Where did this come from? Is it an absolute biomass of the whole population, just the spawning population, or something else?

This text was added in lines 104-106: It is biomass of the adult population, since the 2011 biomass estimate was made in the fish reproductive aggregations areas of croaker. Like the catch data, it only includes the adult population.

Related to the above question, there is an implicit assumption of what biomass is (and defined through the catch selectivity curve), but it is not made explicit. Please given some detail as to what the B value includes, as I am not sure it is total biomass.

The answer to this question is already addressed in the previous answer. The referee asked is the biomass of the population estimated in 2011 and that, is the total of the entire population. It is already clarified that is the biomass of the adult population

The stock reduction in 2011 is based on what? Is it expert opinion?

I do not sure about this question. There are two mentions of biomass from 2011; first, the one estimated by Arzola et al 2018 by the swept area method and if referee refers to that, is an estimate that we do not made. Second, the estimated biomass series that we put are precisely the result of the SRA. It is not expert opinion; it is an estimate.

Instead of just citing a reference, maybe say a bit more where the natural mortality value comes from. Is it directly measured or empirically derived?

This text was added in lines 103-104: M values were determined from six empirical equations

Are catches being used total removals from all fisheries? Is there any dead bycatch not being considered? Anyone of these things can mess us the estimation. Lines 235-237 provide some insight here, and provides what seems to be additional results. Why are those not in the main Methods and Results section? Those results are as important as any presented. I highly recommend incorporating the exploration of uncertainty in catches into the main Methods and Results sections, not just in the Discussion.

In this case we add a text in method and results sections to clarify the unrecorded catch that in previous version was only in the discussion section.

METHODS: (lines 123-125)

To explore the effect of unreported catch in the official record, the original catch was increased by 20% and 40% and then the SRA was applied assuming M = 0.51 and restricting the solution to a B2011 = 14412.

RESULTS: (lines 177-185)

The effect of a constant unreported catch rate of 20 and 40% was a 3% and 6% reduction in the rate of decline of biomass respectively (Table 3). on the other hand, biomass increased by 7% and 11% and recollecting also increased by 6% and 12%. But it is fair to mention that in the 40% catch compensation scenario, the acceptable solution was at the limit. A higher rate of underreporting is not an acceptable solution if B2011 = 4412 tons.

Table 3. SRA results under different unreported catch rate scenarios assuming M = 0.51 and B2011 = 14412 tons.

Setting (%)

P

B1

R

B2011

0

0.74

22,186

8,869

14,410

20

0.71

23,729

9,478

14,418

40

0.67

24,797

10.039

14,418

DISCUSSION: (lines 270-294)

Shift this text:

The effect of this unrecorded catch is an underestimation of the actual size of the stock and an overestimation of the proportion of stock reduction throughout the series. In this work, it was observed that increasing 20% or 40% of the original catch to compensate for the unrecorded catch resulted in an increase in the same proportion of B1, R, B2011 in the SRA, while the reduction proportion of the Stock only decreased at the rate of 0.00015 for each percentage unit of catch increased to compensate for the unrecorded catch.  Thus, with an unrecorded catch of 20%, P decreased 0.03 percentage points (e.g. from 0.73 to 0.70) and considering a 40% underreported catch, P decreased 0.06 percentage points (e.g. 0.73 to 0.67) ). Considering the lowest value of P = 0.62 estimated with the original data, assuming an extreme of 40% of unrecorded catch, when correcting the catch, the SRA would yield a P = 0.56 and does not change the conclusion of the condition of the stock. However, it could also be seen that by compensating a 40% of unreported catch, the minimum acceptable solution was just below the B2011 mean of 14,412 tons estimated per swept area, so it is considered that if there is a percentage of catch not registered in this fishery this must be less than 40%.

With This text:

The exploration of unrecorded catch scenarios indicates that the effect translates into an overestimation of the rate of decline of biomass (P) and an underestimation of the initial biomass and Recruitment. However, the biomass reduction rate, which indicates the condition of the stock, was reduced by just 0.07 if an extreme unreported catch of 40% is assumed. Considering the lowest value of P = 0.62 estimated with the original data, assuming an extreme of 40% of unrecorded catch, when correcting the catch, the SRA would yield a P = 0.56 and does not change the conclusion of the condition of the stock. However, it could also be seen that by compensating a 40% of unreported catch, the minimum acceptable solution was at the limit, suggesting that the underreporting in this fishery should be less than 40%.

Minor considerations

Lines 97-99: More description of what you mean by “the expected recruitment line was used” is needed. I think you mean to say that at the beginning of the time series the population is at equilibrium, thus recruitment must offset death. More detail will help the reader understand the many assumptions inherent in an SRA.

The text in red was included in the main text in lines 108-109 of new version

The Conclusions section is odd. It just presents conclusions from the hyperstability analysis. As this was just a small part of the overall paper, why is it the only thing in the Conclusions section?

This text was added in line 336-341 of new version: According to the results the bigeye croaker stock in the upper Gulf of California keeps underexploited level. Irrespectively of the previous conclusion and considering that the species reproductive behavior is to travel long distances to spawn in aggregations, that persist for approximately three months, the fishery must be managed with extremely caution to keep it suitable.

Edits/Suggestions:

Lines 36-42: This is a run on sentence. Please break this up into 2-3 sentences.

Done: see line 36 of new version

Lines 43-44: “There is scarce information about growth and mortality, and only one estimate of biomass from 2011.”

Done: see lines 44-45 of new version

Line 45: “… knowing the status of the stock. Measuring stock status is difficult because…”

We add the suggested text see line 47 of new version

Line 68: ¨Pesca,,- why the quotes and double commas? You could also end the sentence here and start the next as “This information is registered…”

The authors thank for observing this mistake of the first version. The quotes and double commas were eliminated. Also, the sentence was split into two sentences see lines 72-73 of new version

Line 186, 272: “species”

Done. We corrected all the mistakes in spelling the of word specie by species (several times through the text).

Line 208: Should “But” be capitalized? It follows a period. I don’t think a comma should be used, as it would create a run-on sentence, so breaking the sentences apart makes sense.

Done as suggested by the reviewer see line 237 of new version

Line 210: “acceptable solution of the acceptable resolution…”—I am not sure what the “acceptable resolution” means. Is this the units of biomass, a time period, something else? It refers to the Baranov catch equation, but it is not clear what it means.

Done: In line 239 of the new version, we erased the sentence

Line 213: What is “RAS”—should this be SRA?

Done the RAS was replaced with SRA. See line 242 of the new version

Lines 214-216: How does greater variation come from restricting confidence intervals? I suspect this is a grammatical error that needs to be fixed.

This text was modified in lines 245-246 of the new version: SRA solutions were most similar between different values of M (CI95%), than between biomass (CI95%) values used as restrictors.

Line 233: Should Bv be B1?

This text was added in lines 263-265 of the new version: Note that when the catch series starts, from the first years of the fishery, the biomass was virgin or with a low exploitation, the replacement line was used and then B1 becomes Bv

Line 233: I do not think it can be said that SRA is reliable. It is what it is, and under many assumptions it can offer some insight into the population dynamics. But to say it is reliable oversells it, as what it is reliable for is undefined here. It happens that the stock status seems above the target even under a variety of scenarios, so it can be said the stock status results are robust to explored uncertainty.

This text was modified in line 266, but complete sentence was writing in lines 265-268 of the new version: The SRA is a simple and effective tool to estimate the condition of the stock in fishery resources with little scientific information, but it depends on a historical series of reliable catches.

Line 262: “Like the…” what? It just gives a reference instead of a name of the method.

Done: see lines 305-306 of the new version